# Sensor-Based Rehabilitation in Neurological Diseases: A Bibliometric Analysis of Research Trends

**DOI:** 10.3390/brainsci13050724

**Published:** 2023-04-26

**Authors:** Salvatore Facciorusso, Stefania Spina, Rajiv Reebye, Andrea Turolla, Rocco Salvatore Calabrò, Pietro Fiore, Andrea Santamato

**Affiliations:** 1Department of Medical and Surgical Specialties and Dentistry, University of Campania “Luigi Vanvitelli”, 80138 Naples, Italy; s.facciorusso89@gmail.com; 2Spasticity and Movement Disorders “ReSTaRt”, Unit Physical Medicine and Rehabilitation Section, Department of Medical and Surgical Sciences, University of Foggia, 71122 Foggia, Italy; andrea.santamato@unifg.it; 3Division of Physical Medicine and Rehabilitation, Faculty of Medicine, University of British Columbia, Vancouver, BC V5Z 2G9, Canada; 4Department of Biomedical and Neuromotor Sciences-DIBINEM, Alma Mater Studiorum Università di Bologna, 40138 Bologna, Italy; 5IRCCS Centro Neurolesi Bonino Pulejo, 98121 Messina, Italy; 6Neurorehabilitation Unit, Institute of Bari, Istituti Clinici Scientifici Maugeri IRCCS, 70124 Bari, Italy

**Keywords:** sensor-based, neurorehabilitation, bibliometric, CiteSpace

## Abstract

Background: As the field of sensor-based rehabilitation continues to expand, it is important to gain a comprehensive understanding of its current research landscape. This study aimed to conduct a bibliometric analysis to identify the most influential authors, institutions, journals, and research areas in this field. Methods: A search of the Web of Science Core Collection was performed using keywords related to sensor-based rehabilitation in neurological diseases. The search results were analyzed with CiteSpace software using bibliometric techniques, including co-authorship analysis, citation analysis, and keyword co-occurrence analysis. Results: Between 2002 and 2022, 1103 papers were published on the topic, with slow growth from 2002 to 2017, followed by a rapid increase from 2018 to 2022. The United States was the most active country, while the Swiss Federal Institute of Technology had the highest number of publications among institutions. *Sensors* published the most papers. The top keywords included rehabilitation, stroke, and recovery. The clusters of keywords comprised machine learning, specific neurological conditions, and sensor-based rehabilitation technologies. Conclusions: This study provides a comprehensive overview of the current state of sensor-based rehabilitation research in neurological diseases, highlighting the most influential authors, journals, and research themes. The findings can help researchers and practitioners to identify emerging trends and opportunities for collaboration and can inform the development of future research directions in this field.

## 1. Introduction

Neurological diseases, such as stroke, traumatic brain injury, spinal cord injury, Parkinson’s disease, and multiple sclerosis, are major public health concerns. Their global burden is significant, with neurological disorders being the leading cause of disability worldwide according to the World Health Organization and affecting over one billion people [1,2].

Contributing 11.6% of the global DALYs (disability-adjusted life years) and 16.5% of the deaths from all causes, neurological disorders remain the leading group cause of DALYs and the second leading group cause of deaths in the world [3]. In addition to the personal and emotional impact of these disorders, they also have a significant economic impact, with the costs of neurological disorders estimated to be over USD 800 billion annually in the USA [4]. 

Governments will have to deal with an increase in the demand for treatment, rehabilitation, and support services for neurological illnesses, as populations are aging, and significantly disabling neurological conditions are more common as people age. Care for these conditions relies heavily on rehabilitation, which aids patients in regaining lost function, enhancing their quality of life, and resuming work and other regular activities.

Despite the significant burden of these disorders, there are currently no established or distinct risk factors to focus on for prevention and treatment, emphasizing the need for additional studies and comprehension to create effective strategies [5]. In recent years, there has been a growing interest in using technology, such as sensors, to improve the assessment and evaluation of patients, as well as the effectiveness and efficiency of rehabilitation [6,7].

Sensors are devices that can detect and respond to physical changes, such as changes in temperature, pressure, or movement [8]. They can be classified into several categories, including mechanical, chemical, optical, and biological sensors [9]. The spread of sensors in medicine has been rapid and widespread, with sensors now being used in a variety of settings, including hospitals, clinics, and even patients’ homes [10]. The use of sensors is expected to continue to grow in the coming years with the development of new and more advanced sensors, as well as the integration of existing sensors into new devices and systems. The development of new technologies and manufacturing techniques has led to the production of sensors at a lower cost, making them more accessible to healthcare providers and patients. This has allowed for the greater implementation of sensor-based technologies in both clinical and home settings, increasing the opportunities for monitoring, diagnosis, and rehabilitation. 

The use of sensors in medicine has evolved significantly over the last few decades. Initially, sensors were used primarily to monitor vital signs such as heart rate, blood pressure, and body temperature [11]. However, with the advancement of technology, sensors have become increasingly sophisticated and are now used for a wide range of applications, including neurorehabilitation [11]. In the field of neurorehabilitation, sensors can be used to measure a wide range of physiological and biomechanical parameters, such as movement, muscle activity, and brain activity, which can provide valuable insight into the functioning of the nervous system [10]. This information can be used to diagnose neurological disorders, track the progress of treatment, and monitor the recovery of patients. The use of sensors in neurorehabilitation also allows for the real-time tracking and monitoring of patients, which can help clinicians to adjust treatment strategies as needed to optimize outcomes [12,13]. Overall, sensor technology has the potential to revolutionize the field of neurorehabilitation, providing new tools for diagnosis, treatment, and monitoring that can improve the lives of patients with neurological disorders [14].

However, the use of sensors can also pose significant challenges in terms of data processing and analysis. The large amount of data generated by sensors can be difficult to analyze, especially when dealing with high-dimensional data sets. This can make it difficult to identify patterns and trends in the data and can also make it difficult to visualize the data in a meaningful way.

Bibliometrics is a quantitative method for analyzing and measuring the scientific literature. It provides a systematic approach to evaluating research outputs and impacts, identifying emerging trends, and mapping the intellectual structure of a specific field [15]. CiteSpace offers a comprehensive suite of clustering and social network analysis methods that allow researchers to identify knowledge gaps, derive novel ideas for investigation, and position their intended contributions to the field. Through its unique visualization capabilities, CiteSpace [16] provides a powerful tool for mapping, generating, and interpreting knowledge maps, allowing users to view the scientific world in new and exciting ways. 

In this study, we conducted a bibliometric analysis on sensor-based rehabilitation in neurological diseases, exploring emerging trends, research collaborations, and the intellectual structure. The aim of this study was to identify the current state of the literature on sensor-based neurorehabilitation, explore the trends and patterns in the research, and find potential areas for future research. 

## 2. Materials and Methods

### 2.1. Data Collection

This bibliometric study used the Web of Science Core Collection (WoSCC) as the source database for data retrieval. The complete data retrieval strategy is illustrated in Figure 1. We did not include some relevant keywords related to sensors, such as “neural electrode” and “brain probe”, in the search strategy because they did not affect the overall results, and we wanted to maintain a clear and consistent search for future comparisons.

A total of 1788 original English articles published between 1 January 2002 and 31 December 2022 were screened. The analysis was conducted on 31 December 2022. In the study, the document types that were analyzed and included in the sample consisted of two categories, namely, “article” and “review”. The inclusion of both types of documents allowed for a comprehensive analysis of the current state of knowledge and understanding in the field. Excluded from the analysis were proceedings papers, book chapters, and articles available only as early access. To obtain data for the analysis, the complete details of each article, such as publication outputs, research categories, authors/co-cited authors, countries/institutions, journals/co-cited journals, co-cited references, and keywords, were obtained from the WoSCC database. The bibliometric analysis procedure was conducted in accordance with best practices and guidelines [17] to ensure the quality of the analysis.

### 2.2. Data Analysis

CiteSpace 6.1.R6 [18] was used to perform bibliometric and visual analyses. This scientometric software also produces visual networks of authors, research categories, countries, institutions, cited journals, keywords, co-citation, and co-occurrence [19,20,21,22]. Microsoft Excel was used to create tables and for the R^2^ trend analysis.

Co-occurrence analysis was used to identify the relationships between words in the documents and measure the frequency of their occurrence to reveal the underlying themes. Co-citation analysis was also used to detect the intellectual structure and emerging sensor-based technology among rehabilitation topics from the selected bibliographic data [23,24]. 

To evaluate the structural quality of the networks and the clustering organization, three structural metrics were used: the average silhouette score [25], the modularity Q index [26], and the betweenness centrality [27] were used to measure the structural quality of the networks. Burstiness [28] was used to detect sudden and significant changes in the frequency of certain features over time, allowing a more comprehensive analysis of entities’ behaviors and impacts.

Cluster labeling was conducted automatically using two algorithms, the Log-Likelihood Ratio (LLR) and the Latent Semantic Indexing (LSI) [29] function, within CiteSpace to compare the occurrences of terms in the citing articles [30].

## 3. Results

### 3.1. Publication Outputs and Time Trend

We found 1103 papers on the topic of sensor-based rehabilitation in neurological diseases. The publications included 1003 articles and 100 reviews. As shown in Figure 2, the annual publications showed slow growth between 2002 and 2017, with 396 publications in 15 years representing 35.9% of the total publications. Between 2018 and 2022, the annual publication output showed a rapid increase, with 707 publications in 5 years, representing 64.1% of the total publications. The number of publications reached a peak in 2021 with 170 publications. The H-index for all publications was 63.

The linear regression analysis showed that the number of publications increased significantly in the examined decades (r-squared = 0.9327; *p*-value < 0.001). Overall, the exponential growth of publications demonstrated an increasing interest in this field of sensor-based rehabilitation.

### 3.2. Hot Topics in Literature Research 

Subject categories were extracted from the Web of Science and mapped with CiteSpace. The generated graph showed 90 nodes, which suggests that the field of study involves 90 categories (Figure 3). The most frequent was rehabilitation (273 distributions), followed by engineering biomedical (246 distributions) and engineering electrical electronic (228 distributions). Other frequent categories included “neuroscience” (188 distributions), “instruments & instrumentations” (176 distributions), “chemistry analytical” (133 distributions), and “clinical neurology” (102 distributions). 

### 3.3. Country Analysis

A total of 74 countries participated in publications on sensor-based rehabilitation in neurological diseases between 2002 and 2022. The top 10 most active countries are shown in Table 1. Table 1 shows the top 10 countries/regions by publication, co-occurrence, and centrality in this field of research. The United States contributed the most papers (303 publications, 27.47%), followed by Italy (137 publications, 12.42%) and China (133 publications, 12.05%). 

Figure 4 illustrates the visual network map of collaborations, demonstrating a high level of cooperation. The highest-ranked country by centrality was the USA (centrality 0.40), followed by Brazil (centrality 0.21) and England (centrality 0.20).

### 3.4. Institution Analysis

A total of 1625 institutions had published articles on sensor-based rehabilitation in neurological diseases in the 20 years analyzed. Table 2 lists the top 10 institutions depending on the number of publications. The three institutions with the highest number of publications were the Swiss Federal Institute of Technology in Switzerland, Oregon Health & Science University in the USA, and Universitat Zurich in Switzerland.

Figure 5 shows the network cooperation map of institutions obtained by using CiteSpace.

### 3.5. Journal Analysis

A total of 348 journals published papers related to sensor-based rehabilitation in neurological diseases between 2002 and 2022. The top 10 most active journals are shown in Table 3. In total, 391 papers were published by these journals, accounting for 35.45% of the total publications. *Sensors* published the most papers (129 papers), accounting for 11.69% of all papers. The highest-ranking journal was the *Journal of Neuroengineering and Rehabilitation* with 65 publications and an impact factor of 5.208, the only journal in the top 10 with an impact factor greater than 5.000. Six journals had an IF between 5.000 and 3.000, while three journals had an IF < 3.000 (minimum 2.356). 

The journals *Archives of Physical Medicine and Rehabilitation*, *Journal of Neuroengineering and Rehabilitation*, and *IEEE Transactions on Neural Systems and Rehabilitation Engineering* are the three most co-cited journals.

### 3.6. Author Analysis 

A total of 4777 authors published papers on sensor-based rehabilitation in neurological diseases. Table 4 shows the top 10 active authors and their related information. They published 100 papers, which accounted for 0.9% of the total number of papers. The top three ranked authors by publication count, with 11 publications each, were Catteneo Davide from Italy, Curt Amin from England, and Horak Fay B from the USA. Of the 10 top authors, 50% were from the USA, while 30% were from Italy.

Figure 6 shows the collaboration network of authors who published on this topic between 2002 and 2022. A visual exploration of authors who published articles on sensor-based rehabilitation in neurological diseases between 2002 and 2022 is shown in Appendix A.

### 3.7. Analysis of References 

A total of 822 references were cited during the 20 years between 2002 and 2022. The 10 top references with the most citations are presented in Table 5.

The top 20 co-cited references with the strongest citation burst can be observed in Figure 7b. Of these, the article with the strongest burst is “A review of wearable sensors and systems with application in rehabilitation” in the *Journal of Neuroengineering and Rehabilitation* published by Patel et al. [9] in 2012. This paper is an important contribution to the field of rehabilitation technology, as it provides a comprehensive review of the developments in wearable technology and its applications in rehabilitation. 

The co-citation cluster map (Figure 7a) revealed 16 clusters with a q value of 0.891 and a silhouette value greater than 0.9. The largest clusters are #0 propulsion, #1 exoskeleton, #2 assessment, and #3 machine learning. The timeline of all co-citation references between 2002 and 2022 is shown in Appendix A.

### 3.8. Keywords Analysis

Keywords may reflect current topics and anticipate forthcoming research boundaries that are more engaging. As shown in Figure 8a, the top three keywords with the highest occurrence were rehabilitation, stroke, and recovery. The top 25 keywords with the strongest burst are indicated in Figure 8b. The timeline of all keywords is shown in Appendix A. Among these keywords, induced movement therapy, accelerometry, multiple sclerosis, and subacute stroke have the strongest burst strength. Moreover, task analysis, stroke, machine learning, and robot sensing system are the most recent burst keywords, indicating that these areas are rapidly evolving and generating significant interest in the scientific community. 

In addition, the keywords could be divided in 11 clusters; a list of the clusters with the main keywords are reported in Table 6. The largest identified cluster is #0 machine learning, which is an important tool for interpreting and analyzing data collected by sensors. Clusters #2, #3, and #8 focused on specific neurological conditions, such as Parkinson’s disease, spinal cord injury, and stroke, and clusters #1, #6, and #7 are related to sensor-based rehabilitation technologies aimed at monitoring movement and improving outcomes.

## 4. Discussion

The present study aimed to conduct a bibliometric analysis of the research on sensor-based rehabilitation in neurological diseases published over the past 20 years. The results showed a steady increase in the number of publications over the years, with a significant increase in the last five years, indicating a growing interest in this field. The exponential growth of publications suggests the potential of this area of research to address the challenges of neurological diseases.

The subject categories extracted from the Web of Science and mapped with CiteSpace showed that the field of study involves 90 categories. Rehabilitation, biomedical engineering, and electrical engineering were the most frequent categories, indicating the interdisciplinary nature of this field. Other categories, such as neuroscience, clinical neurology, and analytical chemistry, also made significant contributions. This diversity in subject categories suggests that research in this area is not only complex but also requires collaboration and interdisciplinary efforts from experts with a wide range of knowledge and skills, highlighting the interdisciplinary nature of the field. It is crucial for researchers to recognize the importance of collaboration and the exchange of knowledge across different fields to address the challenges and improve the outcomes of sensor-based rehabilitation interventions. 

The analysis of the countries with the highest number of publications showed that the United States, Italy, and China were the most active countries. This indicates that research on sensor-based rehabilitation in neurological diseases is a global concern, with active participation from multiple countries. The visual network map of collaborations demonstrated a moderate degree of cooperation among the countries.

The institutions with the highest number of publications included the Swiss Federal Institute of Technology in Switzerland, Oregon Health & Science University in the USA, and Universitat Zurich in Switzerland. It is noteworthy that the number of institutions involved in sensor-based rehabilitation research in neurological diseases is growing, indicating the increasing interest in and recognition of this field. The inclusion of new institutions in this research area can contribute to the development of new ideas and approaches, as well as promote the dissemination of knowledge and collaboration between different research groups. Thus, it is important to continue tracking the trends and changes in the institutional landscape of this field in order to better understand its evolution and potential future directions.

The main active authors identified in this study were from the United States and Italy. This suggests the significant contributions made by these authors to the field of sensor-based rehabilitation in neurological diseases. The fact that the number of publications per author in this discipline was very modest is also significant to notice because it suggests that there are not any dominant or extraordinarily prolific authors in this field. This highlights the need for continued collaboration and interdisciplinary research efforts to advance the field of sensor-based rehabilitation in neurological diseases. The lack of a few standout authors also suggests that progress in this field is the result of the collective efforts of numerous researchers, rather than the work of a small group of individuals. Moreover, the analysis of author and institutional collaborations in this study has revealed a notable lack of strong research groups within the network. According to this result, there is a lot that can be done to create more coherent research groups that span several disciplines. It may be required to prioritize multidisciplinary collaboration and bring together academics with various backgrounds and skill sets in order to form more productive research groups. This can entail dismantling conventional disciplinary boundaries, encouraging interdisciplinary cooperation, and establishing a climate of open communication and knowledge sharing. Additionally, it could be crucial to set up specific research goals and objectives as well as to make sure that everyone in the group is on the same page with regard to the research’s approach and overall vision.

The analysis of journals publishing papers related to sensor-based rehabilitation in neurological diseases showed that *Sensors* published the most papers, while the *Journal of Neuroengineering and Rehabilitation* had the highest impact factor. The analysis also identified the three most co-cited journals, namely, *Archives of Physical Medicine and Rehabilitation*, *Journal of Neuroengineering and Rehabilitation*, and *IEEE Transactions on Neural Systems and Rehabilitation Engineering*.

A stronger knowledge of the trends in a particular study discipline is provided through references with citation bursts. We chose to analyze the references with the strongest citation bursts because they are indicative of publications that have garnered significant attention within the scientific community. These publications are likely to contain groundbreaking research or ideas that have had a significant impact on the field of sensor-based rehabilitation in neurological diseases. 

The paper by Patel et al. revealed the strongest burst, lasting 4 years: the paper *A review of wearable sensors and systems with application in rehabilitation*, published in the *Journal of Neuroengineering Rehabilitation* in 2012 [9], is a review that summarizes recent developments in wearable sensors and systems relevant to rehabilitation. It highlights the growing use of wearable technology to monitor older adults and individuals with chronic conditions in home and community settings. The paper describes enabling technologies, such as sensor and communication technologies and data analysis techniques, that allow for the implementation of wearable systems. The review focuses on the clinical applications of wearable technologies, including health and wellness, safety, home rehabilitation, the evaluation of treatment efficacy, and the early detection of disorders. It also discusses the integration of wearable and ambient sensors for home monitoring and outlines future work needed for the clinical deployment of wearable sensors and systems.

Baily et al. revealed the second-strongest citation burst with the paper *Quantifying Real-World Upper-Limb Activity in Nondisabled Adults and Adults with Chronic Stroke*, published in *Neurorehabilitation and Neural Repair* in 2015 [36]. The study aimed to quantify real-world bilateral upper-limb activity in nondisabled adults and adults with chronic stroke using an accelerometry-based methodology. Nondisabled adults demonstrated the equivalent use of dominant and nondominant upper limbs, whereas adults with stroke showed lower bilateral upper-limb activity intensity and more lateralized activity. The study concluded that the novel accelerometry-based methodology can complement clinical tests of function when assessing the recovery of upper-limb activity following neurological injury.

On the other hand, we also chose to examine the references with the most recent bursts because they are indicative of emerging trends or current areas of active research in the field. By analyzing these references, we can gain insights into the most recent advancements and directions of research within sensor-based rehabilitation for neurological diseases. *Wearable Movement Sensors for Rehabilitation: A Focused Review of Technological and Clinical Advances* by Porciuncula et al. [50] is a review of the applications of wearable movement sensors in rehabilitation. The article presents state-of-the-art and next-generation wearable movement sensors, as well as clinical applications in various conditions, including stroke, movement disorders, knee osteoarthritis, and running injuries. The potential for remote monitoring, telerehabilitation, and robotics is also discussed, along with complementary applications enabled by next-generation sensors. Overall, the review highlights the tremendous potential of wearable sensors to change clinical practice and enable personalized and precision medicine.

The other two papers focus on the use of wearable sensor networks for stroke patients. The first paper, *A remote quantitative Fugl-Meyer assessment framework for stroke patients based on wearable sensor networks* by Yu et al. [44], proposes a remote quantitative Fugl-Meyer assessment framework for stroke patients that uses wearable sensors to monitor the movement function of the upper limb, wrist, and fingers. The second paper, *Exploring the Role of Accelerometers in the Measurement of Real World Upper-Limb Use After Stroke* by Hayward et al. [51], explores the feasibility of using accelerometers as a tool to measure real-world upper-limb use after stroke and discusses the potential for widespread uptake in research and clinical environments. Both papers highlight the potential of wearable sensor networks to enhance stroke rehabilitation and evaluation, as well as the need for consistent protocols, applications, and data interpretation to facilitate greater uptake.

Subsequently, we analyzed the 20 references exhibiting the highest citation bursts. Based on our analysis, we found that the majority of references with the strongest citation bursts (13 out of 20) were related to stroke rehabilitation. This suggests that sensor-based rehabilitation is a highly researched area in stroke rehabilitation, with a significant amount of literature available on the topic. Furthermore, 50% (10 out of 20) of the references specifically focused on upper-limb rehabilitation, indicating a strong emphasis on upper-limb rehabilitation after stroke. Finally, we found that 7 out of 20 were review or systematic review papers, highlighting the importance of synthesizing and summarizing the existing literature in this area. 

An important theme that emerged from the analysis of the references with the strongest citation bursts is the use of Kinect technology in sensor-based rehabilitation [33,40,42]. The Kinect system, developed by Microsoft for gaming, has been explored for its potential to monitor and improve motor function in various patient populations. Its advantages include its low cost, ease of use, and ability to capture three-dimensional motion data without the need for cumbersome sensor placement. Additionally, the Kinect system allows for an objective and quantitative assessment of movement, which can be particularly useful in tracking progress during rehabilitation [52,53]. However, some studies have also highlighted limitations of the Kinect system, such as its sensitivity to environmental factors and potential accuracy issues in capturing complex movements [54]. 

The analysis of keywords in the bibliometric study of sensor-based technology in rehabilitation for neurological diseases revealed several themes and trends in this field. The most commonly used terms were related to the core concept of rehabilitation, stroke, and recovery, indicating the strong focus on the use of technology to aid in the recovery process of individuals with neurological conditions. Other terms, such as Parkinson’s disease, spinal cord injury, cerebral palsy, multiple sclerosis, and traumatic brain injury, demonstrated the wide scope of neurological diseases being investigated in this field. The importance of walking and gait analysis was also highly prioritized, reflecting the need for accurate and precise measurements in this area of research. The presence of keywords such as “wearable sensors,” “virtual reality,” and “exoskeleton” indicates a growing interest in using technology to monitor and improve mobility and physical activity. 

The top burst keywords showed emerging topics such as “machine learning”, “deep learning”, “task analysis”, “robot sensing system”, and “brain computer interface”, suggesting that these areas are rapidly evolving and likely to drive new advances in the field and may lead to improved rehabilitation strategies and outcomes for patients and respond to changes in the environment. In particular:Machine learning: Machine learning techniques have become increasingly essential in the analysis and interpretation of data acquired through sensors. These techniques enable the identification of patterns and relationships within the data, which may be difficult or impossible for humans to discern. As a result, machine learning has become a crucial tool in many areas of research, including wearable sensor technology, virtual reality, rehabilitation robotics, and clinical trials, among others [55]. The application of machine learning algorithms allows for the development of more accurate and efficient methods for data processing, which can lead to improved diagnostic and therapeutic approaches [56,57]. Deep learning analysis is a type of machine learning that involves training artificial neural networks to recognize patterns in data. It is a subset of artificial intelligence that uses algorithms to learn from large amounts of data and make predictions or decisions without being explicitly programmed [14]. There are several deep learning models that can be used in rehabilitation, depending on the specific task and type of data being analyzed [58].Task analysis: Task analysis is a critical aspect of rehabilitation that involves breaking down a specific activity or task into its individual components to understand the physical requirements and limitations of the patient. This analysis helps therapists design personalized therapy programs that target the patient’s specific needs and goals [59]. The application of sensors in task analysis can significantly enhance the accuracy and effectiveness of the rehabilitation process. Sensors can provide real-time data on the patient’s movements, muscle activity, and other physical parameters, allowing therapists to identify areas of weakness and adjust the therapy program accordingly.Robot sensing system: A robot sensing system is a set of sensors integrated into a robot to collect data about the robot’s environment and the physical parameters related to its operation. These sensors can include cameras, microphones, force sensors, accelerometers, and others, depending on the specific application of the robot. The data collected by the sensors is used to control the robot’s movements, adjust its behavior, and make decisions based on the information gathered. A well-designed robot sensing system can significantly improve the robot’s functionality, safety, and performance in various applications, such as manufacturing, healthcare, and exploration [60,61,62,63].Brain–computer interface (BCI): BCI is a technology that allows direct communication between the brain’s electrical activity and an external device, most commonly a computer [64]. BCIs can be used in stroke rehabilitation [65,66], as well as in a completely locked-in state to enable volitional communication, allowing patients to select letters, to form words and phrases, and to communicate their needs and experiences via auditory neurofeedback training [67]. BCIs can also be used to control prosthetic devices, such as robotic arms or legs, by translating neural signals into motor commands [68]. Sensors play a crucial role in brain–computer interface (BCI) technology, as they are used to detect changes in brain activity associated with specific mental states or movements. There are different types of sensors that can be used in BCIs, including invasive and non-invasive approaches. Non-invasive sensors include electroencephalography (EEG), magnetoencephalography (MEG), and functional near-infrared spectroscopy (fNIRS). Invasive sensors include microelectrode arrays and penetrating electrodes [69].

The clustering of keywords in this analysis also suggests that there are specific areas of focus within the field, such as Parkinson’s disease, spinal cord injury, and stroke. The largest cluster is focused on machine learning, which is a promising area of research in rehabilitation, as it can be used to identify patterns and trends in patient data and provide personalized feedback to improve outcomes. The continued growth and advancement of machine learning techniques are expected to contribute significantly to the development of new and innovative solutions in the field of biomedical engineering. Human–computer interaction and learning algorithms suggest that researchers are exploring ways to improve the interaction between users and sensor-based systems. These clusters are important because they emphasize the need to design systems that are intuitive and user-friendly for patients affected by neurological disorders.

This study has certain limitations that are inherent to bibliometric analysis. First, the data were collected only from the WoSCC database, which may have missed some relevant studies that are present in other databases. Second, only studies in English were included. Third, some recent high-quality works may not have received enough citations, which could lead to an underestimation of their influence. Although the WoSCC database is regularly updated and citation-specific parameters are subject to temporal changes, the influence of time on citation trends is limited.

Moreover, the data presented in this paper have limitations that preclude their use for some purposes. The metrics we employed only measure the frequency and performance, not their quality or impact. For example, articles may receive numerous citations for negative reasons, such as being refuted or criticized. Additionally, metrics may distort research by incentivizing researchers to produce papers that are more likely to be cited, rather than those that advance knowledge. 

Finally, we acknowledge that a bibliometric analysis has its limitations and cannot account for all the nuances and emerging areas of a scientific field. Therefore, our results should be interpreted with caution and not as a definitive representation of the state of the art in the field. Rather, we suggest that our results provide a useful overview of the main trends and topics in the literature, as well as some insights into the gaps and challenges that need further investigation. To overcome some of the limitations of bibliometric analysis, we recommend that future studies complement our approach with other methods, such as the qualitative analysis of the content and context of selected publications, expert interviews to elicit opinions, and perspectives from key stakeholders.

## 5. Conclusions

The present study provides a comprehensive overview of the research on sensor-based rehabilitation in neurological diseases in the last 20 years. The analysis of journals publishing related papers and references with citation bursts provided insights into the most impactful and groundbreaking research in the field. Overall, this study sheds light on the evolution and potential future directions of the field of sensor-based rehabilitation in neurological diseases. This article contributes to advancing the field of sensor-based rehabilitation and helps to pave the way for future research in this area.

Sensor-based rehabilitation is not only a promising research area but also a rapidly growing market that attracts many established and emerging technologies. According to a recent report by Grand View Research, the global wearable medical device (including sensors) market size was valued at USD 21.3 billion in 2021 and is expected to grow at a compound annual growth rate (CAGR) of 28.1% from 2022 to 2030 [70]. Some of the key factors driving this growth are the increasing prevalence of neurological disorders such as stroke, Parkinson’s disease, multiple sclerosis, and traumatic brain injury; the rising demand for home-based and personalized rehabilitation solutions; and technological advancements in sensor devices, wearable systems, robotics, virtual reality, and artificial intelligence. 

In the context of targeted neural rehabilitation [71], we can expect to see continued advancements in sensor technology that will enable even more detailed and accurate data collection on patient movements, brain activity, and vital signs. These data will be analyzed using machine learning and deep learning algorithms to identify patterns and trends that can inform personalized treatment plans for patients. BCIs are also expected to become more prevalent in sensor-based rehabilitation, allowing for direct communication between a patient’s brain activity and an external device, such as a computer or robotic arm. Looking ahead, virtual reality and gamification are expected to become more prevalent in sensor-based rehabilitation, providing patients with a more engaging and immersive therapy experience. The integration of sensors with other technologies, such as robotics and exoskeletons, is also expected to drive further innovation in this field, leading to the development of more advanced rehabilitation devices that can provide patients with greater levels of assistance and support during therapy sessions. By exploring these emerging technologies and advancements in sensor-based rehabilitation, we can identify new opportunities for innovation and collaboration that will help advance the field and pave the way for future research in this area.

## Figures and Tables

**Figure 1 brainsci-13-00724-f001:**
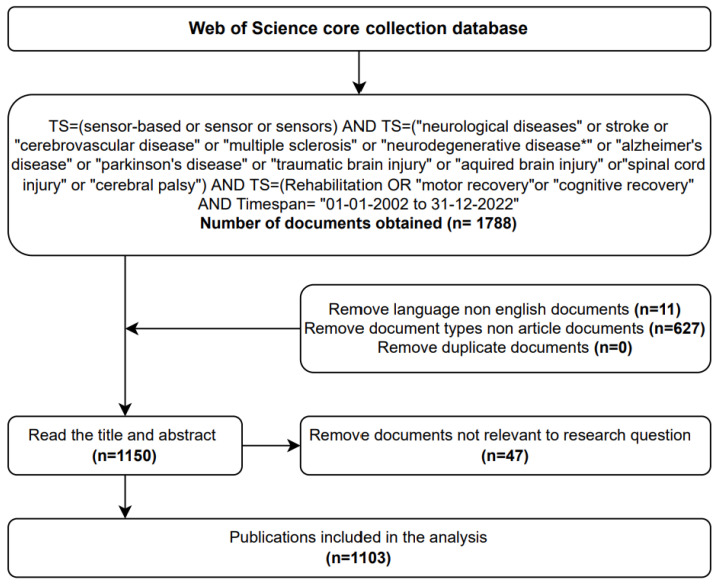
Flowchart of study identification and selection.TS: topic search; (*): any group of characters, including no character.

**Figure 2 brainsci-13-00724-f002:**
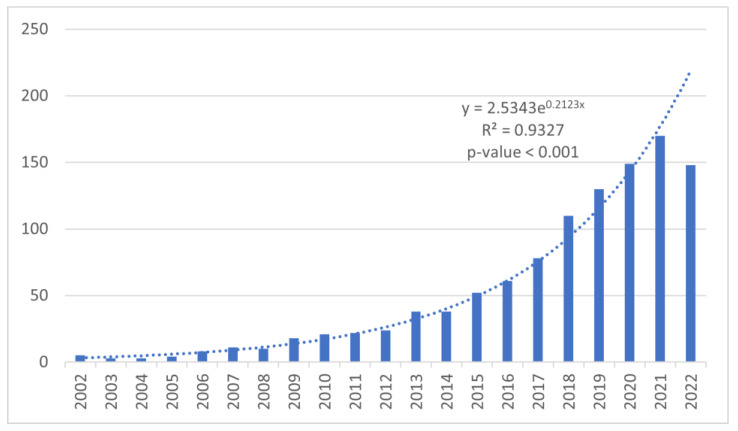
Annual publication trend from 2002 to 2022.

**Figure 3 brainsci-13-00724-f003:**
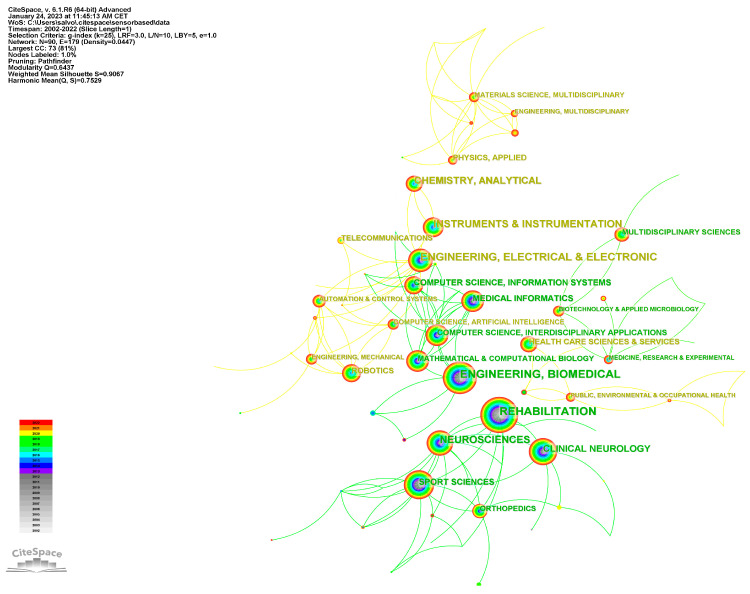
Research hotspot distribution.

**Figure 4 brainsci-13-00724-f004:**
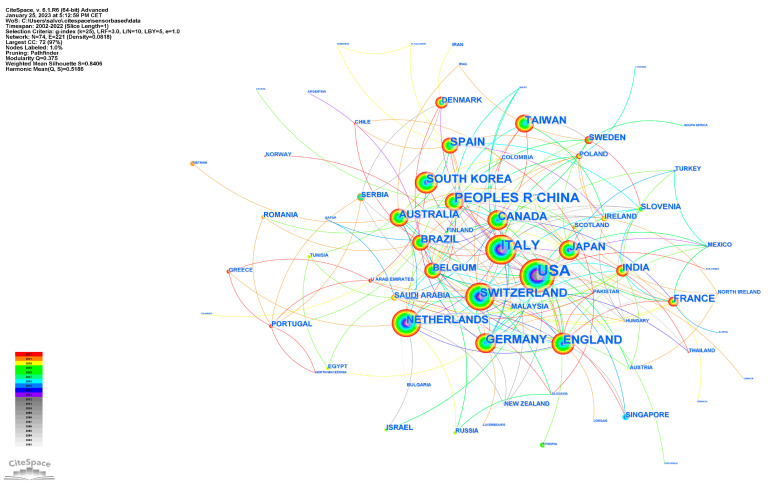
Country network visualization: co-occurrence analysis of countries.

**Figure 5 brainsci-13-00724-f005:**
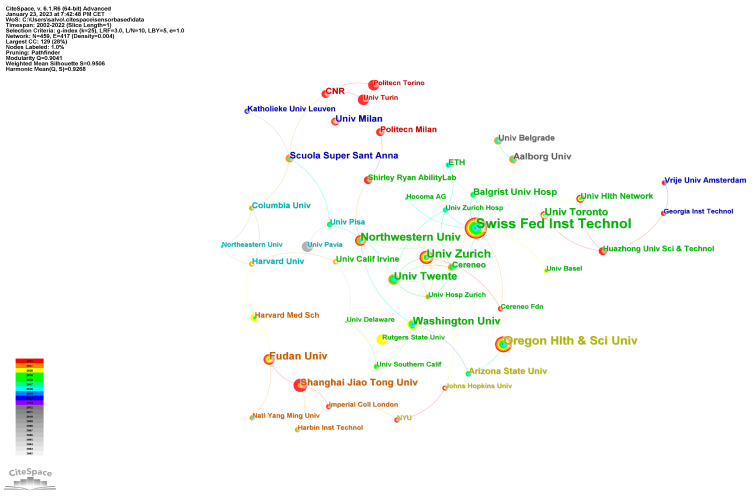
The network cooperation map of institutions.

**Figure 6 brainsci-13-00724-f006:**
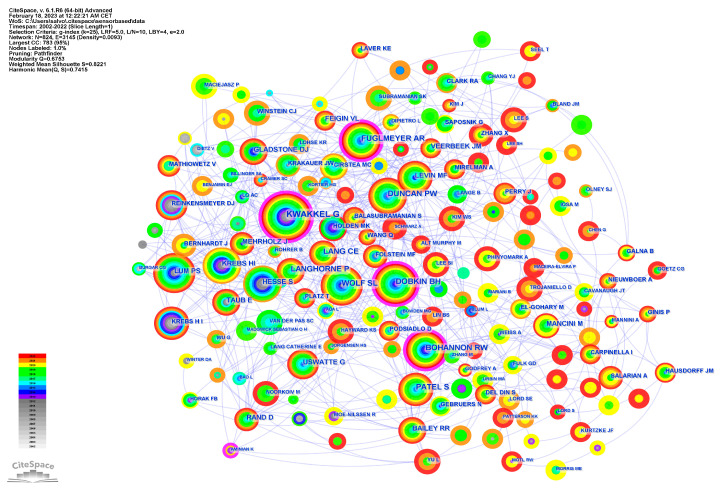
Collaboration network of authors.

**Figure 7 brainsci-13-00724-f007:**
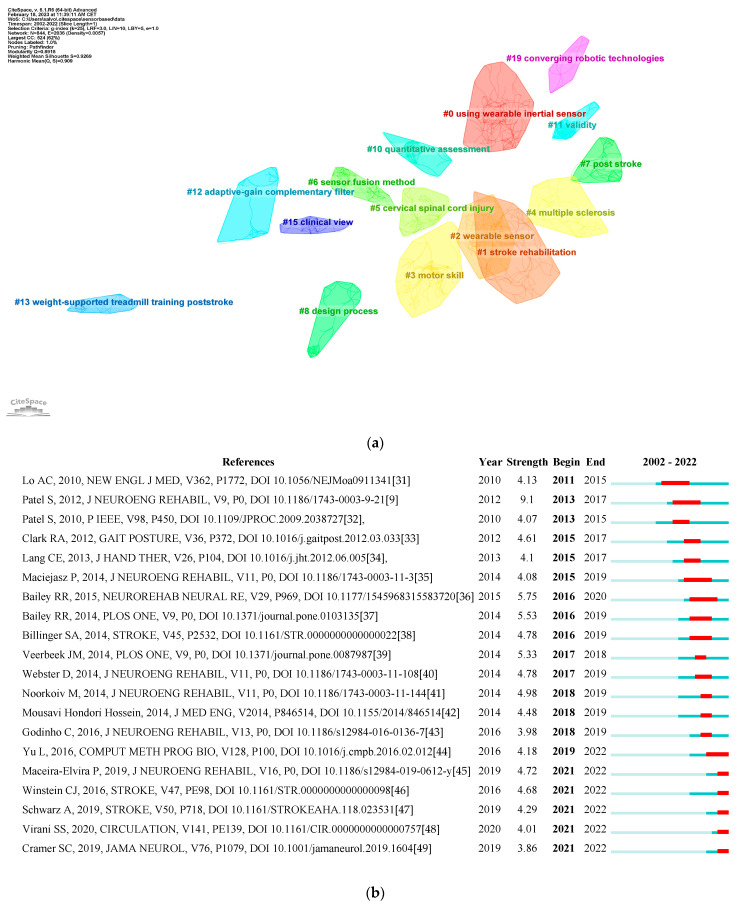
Reference co-citation analysis [31,32,33,34,35,36,37,38,39,40,41,42,43,44,45,46,47,48,49]. (**a**) Cluster map; (**b**) bursts.

**Figure 8 brainsci-13-00724-f008:**
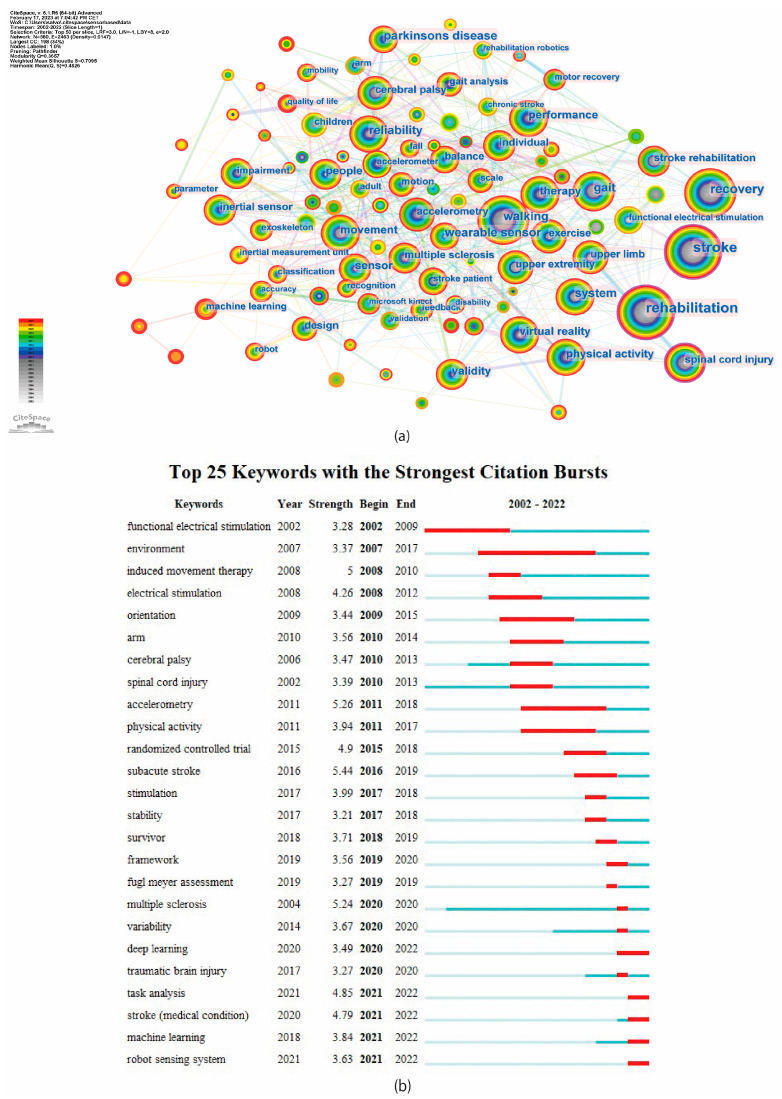
Keywords analysis. (**a**) Co-occurrence counts; (**b**) bursts.

**Table 1 brainsci-13-00724-t001:** The top 10 countries stratified by publication, centrality, and co-occurrence.

Rank	Country Region	Publications	Country Region	Centrality	Country Region	Co-Occurrence
1	USA	303	USA	0.40	USA	294
2	Italy	137	Brazil	0.21	Italy	134
3	China	133	England	0.20	China	132
4	England	71	Italy	0.19	England	66
5	Germany	65	Spain	0.18	Switzerland	65
6	Switzerland	65	Belgium	0.18	Germany	64
7	Canada	58	China	0.16	Canada	55
8	South Korea	54	Australia	0.16	South Korea	52
9	Spain	53	Canada	0.15	Spain	50
10	Netherlands	50	Germany	0.14	Netherlands	49

**Table 2 brainsci-13-00724-t002:** The top 10 most productive institutions.

Ranking	Institution	Country	Publications	Total Link Strength
1	Swiss Federal Institute of Technology	Switzerland	29	8076
2	Oregon Health & Science University	USA	20	2249
3	Universitat Zurich	Switzerland	18	6292
4	Northwestern University	USA	16	5618
5	University of Toronto	Canada	16	2139
6	Fudan University	China	15	2803
7	Washington Univ	USA	14	7816
8	University of Twente	Netherlands	14	3636
9	Case Western Reserve University	USA	14	439
10	Scuola Superiore Sant Anna	Italy	10	2151

**Table 3 brainsci-13-00724-t003:** The top 10 journals and cited journals published on sensor-based rehabilitation in neurological diseases between 2002 and 2022.

Rank	Journal	P	IF	Co-Cited Journal	Cit	IF
**1**	*Sensors*	129	3.847	*Archives of Physical Medicine and Rehabilitation*	612	4.060
**2**	*Journal of Neuroengineering and Rehabilitation*	65	5.208	*Journal of Neuroengineering and Rehabilitation*	571	5.208
**3**	*IEEE Transactions on Neural Systems and Rehabilitation Engineering*	49	4.528	*IEEE Transactions on Neural Systems and Rehabilitation Engineering*	457	4.528
**4**	*Archives of Physical Medicine and Rehabilitation*	27	4.060	*Neurorehabilitation And Neural Repair*	433	4.895
**5**	*Frontiers in Neurology*	27	4.086	*Stroke*	428	10.170
**6**	*IEEE Access*	24	3.476	*Biosensors—Basel*	414	5.743
**7**	*IEEE Sensors Journal*	20	4.325	*Gait & Posture*	412	2.746
**8**	*Applied Sciences Basel*	17	2.838	*Physical Therapy*	495	3.140
**9**	*Gait & Posture*	17	2.746	*PloS One*	358	3.752
**10**	*Medical Engineering & Physics*	16	2.356	*IEEE Transactions on Biomedical Engineering*	311	4.756

IF: impact factor.

**Table 4 brainsci-13-00724-t004:** The top 10 most productive authors.

Rank	Authors	Country	Institution	P	H-Index
1	Cattaneo Davide	Italy	IRCCS Fondazione Don Carlo Gnocchi Onlus	11	27
2	Curt Armin	England	University of Cambridge	11	62
3	Horak Fay B.	USA	Oregon Health & Science University	11	93
4	Ferrarin Murizio	Italy	IRCCS Fondazione Don Carlo Gnocchi, ONLUS	10	160
5	King Laurie A.	USA	Oregon Health & Science University	10	23
6	Lang Catherine E.	USA	Washington University (WUSTL)	10	47
7	Luft Andreas	Switzerland	University Zurich Hospital	10	36
8	Audu Musa L	USA	Case Western Reserve University	9	18
9	Carpinella Ilaria	Italy	IRCCS Fondazione Don Carlo Gnocchi, ONLUS	9	18
10	Dobkin Bruce H	USA	David Geffen School of Medicine at UCLA University of California System	9	57

**Table 5 brainsci-13-00724-t005:** The top 10 most cited references.

Rank	Title	Cit	First author	Journal	Publication Year
1	Soft robotic glove for combined assistance and at-home rehabilitation	803	Polygerinos, P.	*Robotics And Autonomous Systems*	2015
2	Spatio-temporal parameters of gait measured by an ambulatory system using miniature gyroscopes	534	Aminian, K.	*Journal Of Biomechanics*	2002
3	Human motion tracking for rehabilitation-A survey	484	Zhou, H.	*Biomedical Signal Processing And Control*	2008
4	Current Hand Exoskeleton Technologies for Rehabilitation and Assistive Engineering	312	Heo, P.	*International Journal of Precision Engineering And Manufacturing*	2012
5	ARMin: a robot for patient-cooperative arm therapy	263	Nef, T.	*Medical & Biological Engineering & Computing*	2007
6	Methods for gait event detection and analysis in ambulatory systems	228	Rueterbories, J.	*Medical Engineering & Physics*	2010
7	Automating arm movement training following severe stroke: Functional exercises with quantitative feedback in a gravity-reduced environment	208	Sanchez, R.	*IEEE Transactions On Neural Systems And Rehabilitation Engineering*	2006
8	Body-worn motion sensors detect balance and gait deficits in people with multiple sclerosis who have normal walking speed	190	Spain, R. I.	*Gait & Posture*	2012
9	The Promise of mHealth: Daily Activity Monitoring and Outcome Assessments by Wearable Sensors	183	Dobkin, B.	*Neurorehabilitation And Neural Repair*	2011
10	Patient-centered activity monitoring in the self-management of chronic health conditions	181	Chiauzzi, E.	*Bmc Medicine*	2015

**Table 6 brainsci-13-00724-t006:** Keyword clusters.

Cluster	Label	Main Keywords	Mean Year
0	machine learning	gesture recognition; human-computer interaction; learning algorithms; gait recognition	2017
1	wearable sensors	gait analysis; inertial measurement units; six-minute walk; simulator sickness	2014
2	Parkinson’s disease	monitoring technologies; exercise intensity; ambulatory systems; gait recognition	2011
3	spinal cord injury	physical activity; brain–computer interfaces; gait rehabilitation; Hammerstein model	2007
4	virtual reality	deep learning; human-computer interaction; biomechanical modeling; optimal control	2015
5	post-stroke rehabilitation	serious games; Microsoft Kinect; electrodermal activity	2014
6	gait analysis	movement; inertial sensor; accelerometry; balance	2011
7	rehabilitation robotics	virtual reality; sensorimotor interaction; movement analysis; human-machine interface	2008
8	stroke rehabilitation	wearable sensors; physical therapy; motor learning; hand rehabilitation	2010
9	upper extremity	machine learning; outcome measures; body-worn sensors	2009
10	clinical trials	stroke rehabilitation; brain tissue regeneration; physical therapy; virtual reality	2016
11	legged locomotion	haptic interfaces; rehabilitation robotics; sensing systems; motion analysis	2017

## Data Availability

The data presented in this study are available within the text.

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
