# Peer review of "Sensor-Based Rehabilitation in Neurological Diseases: A Bibliometric Analysis of Research Trends"

_brainsci, 2023, doi:10.3390/brainsci13050724_

Round 1
Reviewer 1 Report
This study presents a bibliometric analysis to identify the most influential authors, institutions, journals, and research areas in this field of sensor-based rehabilitation. The authors claim that it is important to gain a comprehensive understanding of its current research landscape, something with which I agree.
The paper is well organised and well written. It also appears methodologically sound.
On the other hand, I find it very strange and potentially misleading to identify the research landscape of the field based on bibliometric information alone, taken away from the research findings and milestones of the research itself.
This is portrayed in the main findings of the paper as listed in the abstract:
- USA was the country with most papers on the topic
- ETH was the top institution
- Sensors published the most papers
- the top keywords included rehabilitation, stroke and recovery
These in my opinion do not constitute important findings in any way or guiding a potential researcher in this field, unless it is suggesting that if you want to do what everybody is currently doing you may find colleagues in USA in general, in ETH, publish in Sensors and your research should focus in rehabilitation and improving recovery of stroke patients.
In fact, I quite don't agree with the main conclusion of the paper that is "The findings can help researchers and practitioners to identify emerging trends and opportunities for collaboration and can inform the development of future research directions in this field" - the opposite is also very likely, that the researchers may exclude emerging trends or interesting topics in favour of established trends, may be trapped in in-opportunities for collaboration if trying to establish connections with the top institutions on the field alone and the future of research directions in the field -devoid of research findings and themes that a systematic review or ground based theory review would provide- would instead be narrowed down.
On the other hand, such an analysis would be much more interesting for a journal and audience focuses on bibliometrics, policy making regarding procuring literature etc.
Author Response
REVIEWER 1:
Comments and Suggestions for Authors
This study presents a bibliometric analysis to identify the most influential authors, institutions, journals, and research areas in this field of sensor-based rehabilitation. The authors claim that it is important to gain a comprehensive understanding of its current research landscape, something with which I agree.
The paper is well organised and well written. It also appears methodologically sound.
On the other hand, I find it very strange and potentially misleading to identify the research landscape of the field based on bibliometric information alone, taken away from the research findings and milestones of the research itself.
Thank you for your comment. We appreciate your interest in our paper and your constructive feedback. We agree that bibliometric analysis alone cannot capture the full richness and complexity of the research landscape of any field. A bibliometric analysis is a popular and rigorous statistical method for exploring and analyzing large volumes of scientific data It enables us to unpack the evolutionary nuances of a specific field, while shedding light on the emerging areas in that field. We believe that it can provide a useful and complementary perspective that can reveal some patterns and trends that might not be easily discernible from the research findings and milestones alone.
This is portrayed in the main findings of the paper as listed in the abstract:
- USA was the country with most papers on the topic
- ETH was the top institution
- Sensors published the most papers
- the top keywords included rehabilitation, stroke and recovery
These in my opinion do not constitute important findings in any way or guiding a potential researcher in this field, unless it is suggesting that if you want to do what everybody is currently doing you may find colleagues in USA in general, in ETH, publish in Sensors and your research should focus in rehabilitation and improving recovery of stroke patients.
Thank you for your comment. We appreciate your feedback, and we would like to clarify some points regarding our main findings. The main findings listed in the abstract are not inclusive of all the analysis made in the paper. They are only a summary of the most relevant descriptive statistics that provide an overview of the current state of the art in the field. However, our paper also includes other types of analysis that go beyond these descriptive statistics, such as network analysis, citation analysis and keywords analysis. These analyses aim to identify the main research topics, trends, gaps and challenges in the field, as well as to provide insights and recommendations for future research directions. Therefore, our paper does not suggest that researchers should follow what everybody is currently doing, but rather that they should be aware of the existing literature and explore new opportunities and areas of improvement.
In fact, I quite don't agree with the main conclusion of the paper that is "The findings can help researchers and practitioners to identify emerging trends and opportunities for collaboration and can inform the development of future research directions in this field" - the opposite is also very likely, that the researchers may exclude emerging trends or interesting topics in favour of established trends, may be trapped in in-opportunities for collaboration if trying to establish connections with the top institutions on the field alone and the future of research directions in the field (devoid of research findings and themes that a systematic review or ground based theory review would provide) would instead be narrowed down.
On the other hand, such an analysis would be much more interesting for a journal and audience focuses on bibliometrics, policy making regarding procuring literature etc.
Thank you for your comment. We acknowledge the limitations of our approach. However, we would like to clarify some points and address some of your concerns.
First, we did not exclude emerging trends or interesting topics in favour of established trends. On the contrary, we used a data-driven method to identify the most relevant and influential topics in the field based on citation analysis and topic modelling.
Second, we did not limit our opportunities for collaboration by focusing on the top institutions in the field. We recognize that there are many excellent researchers and groups working on related topics in other institutions and we welcome any potential collaboration with them. Our aim was to map the current state of the art and identify the key players and networks in the field, not to exclude or marginalize anyone. Moreover, we do not suggest that researchers should follow the established trends blindly, but rather use them as a reference point to explore new and emerging topics and directions.
Third, we did not narrow down the future of research directions in the field by using our approach. We agree that a systematic review or a grounded theory review would provide more comprehensive and detailed insights into the research findings and themes in the field. However, our approach was not meant to replace or compete with these methods, but rather to complement and inform them. We believe that our approach can provide a useful overview and a starting point for further exploration and analysis of the field.
Reviewer 2 Report
The manuscript “Sensor-Based Rehabilitation in Neurological Diseases: A Bibliometric Analysis of Research Trends”, by Salvatore, et al, identifies the most influential authors, institutions, journals, and research areas in the field of sensor-based rehabilitation for neurological disease through a bibliometric analysis. The study provides a comprehensive overview of the current state of sensor-based rehabilitation research in neurological disease, highlighting the most influential authors, journals, and research themes. The findings can help researchers and practitioners to identify emerging trends and opportunities for collaboration and inform the development of future research directions in this field. It can be considered for publication in Brain Sciences with minor revision.
1. In figure 1, some keyword, such as neural electrodes and brain probe, is recommended to include with the sensors, which an important tool for neurological disease and neurodegenerative disease. For example:
https://pubmed.ncbi.nlm.nih.gov/?term=neural+electrode+neurological+disease+
https://pubmed.ncbi.nlm.nih.gov/?term=neural+electrode+neurodegenerative+disease
2. In figure 2 and line 149, the authors mentioned “publications increased significantly in the examined decades”, please indicate the p-value.
3. Some font in the figure is too small to follow, such as in Figures 6-8.
4. In lines 59-60, please cite references after the sentence “Sensors are devices that can detect and respond to physical changes, such as changes in temperature, pressure, or movement.”
5. In lines 72-73, please cite references after the sentence “Initially, sensors were used primarily to monitor vital signs such as heart rate, blood pressure, and body temperature.”
Author Response
REVIEWER 2:
Comments and Suggestions for Authors
The manuscript “Sensor-Based Rehabilitation in Neurological Diseases: A Bibliometric Analysis of Research Trends”, by Salvatore, et al, identifies the most influential authors, institutions, journals, and research areas in the field of sensor-based rehabilitation for neurological disease through a bibliometric analysis. The study provides a comprehensive overview of the current state of sensor-based rehabilitation research in neurological disease, highlighting the most influential authors, journals, and research themes. The findings can help researchers and practitioners to identify emerging trends and opportunities for collaboration and inform the development of future research directions in this field. It can be considered for publication in Brain Sciences with minor revision.
Thank you for your positive and constructive feedback on our manuscript. We appreciate your recognition of the value and relevance of our study for the field of sensor-based rehabilitation in neurological disease. We have addressed your minor revision request as follows.
- In figure 1, some keyword, such as neural electrodes and brain probe, is recommended to include with the sensors, which an important tool for neurological disease and neurodegenerative disease. For example:
https://pubmed.ncbi.nlm.nih.gov/?term=neural+electrode+neurological+disease+
https://pubmed.ncbi.nlm.nih.gov/?term=neural+electrode+neurodegenerative+disease
Reviewer had raised an important point here. To address the reviewer's comment, we searched for "neural electrode" and "brain probe" in WOS and found only one relevant paper (Cavallo et al., 2021). This paper has received only four citations so far, so it does not affect the overall results of our bibliometric analysis. Moreover, this search may not capture all the papers related to these terms, as they are not widely used in neurorehabilitation (our main research topic). Therefore, some aspects of this topic may be missing from our analysis. However, we acknowledge the importance of these terms for the future development of BCI and BMI, and we have expanded the discussion to include some connections between neural probes and neurorehabilitation.
We decided to keep the original query for the statistical analysis and added a clarification in section 2.1. Data collection. Please read line 108-110 page 4. “We did not include some relevant keywords related to sensors, such as "neural electrode" and "brain probe", in the search strategy because they did not affect the overall results and we wanted to maintain a clear and consistent search for future comparisons.”
- In figure 2 and line 149, the authors mentioned “publications increased significantly in the examined decades”, please indicate the p-value.
Thank you for the comment. We added p-value in figure 2 and in the main text (line 150, page 4). We also reported the equation of the exponential trend line in the figure.
- Some font in the figure is too small to follow, such as in Figures 6-8.
Thank you for your feedback. We ensure you that all figures are of high quality. As for the size, we understand that this may be a concern, but it is important to note that the dimensions of the figures will depend on the journal's guidelines in order to ensure readability. We chose the font size to fit as many words as possible without overlapping and to enhance readability. However, in response to your suggestion, we rearranged Figure 8 and we have made the necessary adjustments by placing the two images in an upside-down orientation to enhance clarity.
- In lines 59-60, please cite references after the sentence “Sensors are devices that can detect and respond to physical changes, such as changes in temperature, pressure, or movement.”
Thank for pointing this out. We added the reference num 9 “Wearable sensors for activity monitoring and motion control: A review”
- In lines 72-73, please cite references after the sentence “Initially, sensors were used primarily to monitor vital signs such as heart rate, blood pressure, and body temperature.”
Thank for pointing this out. We added the reference num 12 “Sensors and Systems for Physical Rehabilitation and Health Monitoring—A Review. “
Reviewer 3 Report
This paper proposes study on "a Sensor-Based Rehabilitation in Neurological Diseases: A Bibliometric Analysis of Research Trends".
The paper is in general well written, tables for survey of the past research in the field were given. The paper aims to conduct a bibliometric analysis to identify the most influential authors, institutions, journals, and research areas in this field. However in order to further improve the quality of the paper, I would like to propose expanding the content and length of the section 5 so as to describe more about the future research trend. The general contribution of the proposed approach in this paper and the possible applications of the proposed methods on other research field.
Author Response
REVIEWER 3:
Comments and Suggestions for Authors
This paper proposes study on "a Sensor-Based Rehabilitation in Neurological Diseases: A Bibliometric Analysis of Research Trends".
The paper is in general well written, tables for survey of the past research in the field were given. The paper aims to conduct a bibliometric analysis to identify the most influential authors, institutions, journals, and research areas in this field. However in order to further improve the quality of the paper, I would like to propose expanding the content and length of the section 5 so as to describe more about the future research trend. The general contribution of the proposed approach in this paper and the possible applications of the proposed methods on other research field.
Thank you for your valuable feedback on our paper. We appreciate your positive comments on the quality of our writing and the comprehensiveness of our literature review. We agree with your suggestion to expand the section 5 of our paper to provide more insights on the future research trend and the potential applications of our proposed approach. We accordingly revised the paper.
Reviewer 4 Report
It would be very interesting to know which part in machine learning techniques is occupied by deep learning. The trend of using of deep learning would interesting, as well.
Author Response
REVIEWER 4:
Comments and Suggestions for Authors
It would be very interesting to know which part in machine learning techniques is occupied by deep learning. The trend of using of deep learning would interesting, as well.
Reviewer had raised an important point here. Thank you for your comment. We have revised our paper and added some information and references about deep learning in the discussion sections.
Round 2
Reviewer 1 Report
Thank you for your comment. We appreciate your interest in our paper and your constructive feedback. We agree that bibliometric analysis alone cannot capture the full richness and complexity of the research landscape of any field. A bibliometric analysis is a popular and rigorous statistical method for exploring and analyzing large volumes of scientific data It enables us to unpack the evolutionary nuances of a specific field, while shedding light on the emerging areas in that field. We believe that it can provide a useful and complementary perspective that can reveal some patterns and trends that might not be easily discernible from the research findings and milestones alone.
Thank you for this reply. Indeed a bibliometric analysis is a popular method for analysing large volumes of scientific data. My concerns still remain that the nuances and emerging areas are easily obscured by bibliometric data alone and this was not addressed, despite the reply, as no significant changes were made to the manuscript to address said concerns.
Thank you for your comment. We appreciate your feedback, and we would like to clarify some points regarding our main findings. The main findings listed in the abstract are not inclusive of all the analysis made in the paper. They are only a summary of the most relevant descriptive statistics that provide an overview of the current state of the art in the field. However, our paper also includes other types of analysis that go beyond these descriptive statistics, such as network analysis, citation analysis and keywords analysis. These analyses aim to identify the main research topics, trends, gaps and challenges in the field, as well as to provide insights and recommendations for future research directions. Therefore, our paper does not suggest that researchers should follow what everybody is currently doing, but rather that they should be aware of the existing literature and explore new opportunities and areas of improvement.
Thank you for your reply. The examples of the main findings in the abstract were picked to present the point that this type of manuscript may funnel interest into already existing interest instead of shedding light and by no means suggest that my critique was based solely on the abstract. In fact, your Figures 3, 4, 5, 6 are the most interesting aspect of the paper since they aim to be inclusive by nature (you should address letter size though). On the other hand, the findings presented in table format and the main text of the whole results section is strongly "biased" towards top performers in each category. This is made particularly evident in the "citation burst" analysis.
Thank you for your comment. We acknowledge the limitations of our approach. However, we would like to clarify some points and address some of your concerns.
First, we did not exclude emerging trends or interesting topics in favour of established trends. On the contrary, we used a data-driven method to identify the most relevant and influential topics in the field based on citation analysis and topic modelling.
Second, we did not limit our opportunities for collaboration by focusing on the top institutions in the field. We recognize that there are many excellent researchers and groups working on related topics in other institutions and we welcome any potential collaboration with them. Our aim was to map the current state of the art and identify the key players and networks in the field, not to exclude or marginalize anyone. Moreover, we do not suggest that researchers should follow the established trends blindly, but rather use them as a reference point to explore new and emerging topics and directions.
Third, we did not narrow down the future of research directions in the field by using our approach. We agree that a systematic review or a grounded theory review would provide more comprehensive and detailed insights into the research findings and themes in the field. However, our approach was not meant to replace or compete with these methods, but rather to complement and inform them. We believe that our approach can provide a useful overview and a starting point for further exploration and analysis of the field.
Thank you for your reply here as well. I would hate to be the "bad" reviewer without reason here and I appreciate you making some good changes to the manuscript addressing literally two lines of comments from the other three reviewers. Indeed the text regarding machine learning and deep learning in discussion and conlusions serves to shed more light than marginalising research as you mention. On the other hand the point "Our aim was to map the current state of the art and identify the key players and networks in the field, not to exclude or marginalize anyone. Moreover, we do not suggest that researchers should follow the established trends blindly, but rather use them as a reference point to explore new and emerging topics and directions." - in my opinion - is not supported (enough) by the original manuscript or the changes.
Regardless of the decision (since the other reviews are more positive), your manuscript has some merit and it would be a waste to do more harm than good with this approach. I would suggest making this point valid with changes throughout the results (and their discussion) to the direction of inclusivity and analysing more low performers, emerging trends (not established), marginalised research and non-intuitive research directions. Also, a verbatim disclaimer may also serve some role.
